# Umbilical Cord Blood-Derived Cell Therapy for Perinatal Brain Injury: A Systematic Review & Meta-Analysis of Preclinical Studies

**DOI:** 10.3390/ijms24054351

**Published:** 2023-02-22

**Authors:** Timothy Nguyen, Elisha Purcell, Madeleine J. Smith, Tayla R. Penny, Madison C. B. Paton, Lindsay Zhou, Graham Jenkin, Suzanne L. Miller, Courtney A. McDonald, Atul Malhotra

**Affiliations:** 1Department of Paediatrics, Monash University, Melbourne, VIC 3168, Australia; 2The Ritchie Centre, Hudson Institute of Medical Research, Melbourne, VIC 3168, Australia; 3Department of Obstetrics and Gynaecology, Monash University, Melbourne, VIC 3168, Australia; 4Cerebral Palsy Alliance Research Institute & Specialty of Child and Adolescent Health, The University of Sydney, Sydney, NSW 2006, Australia; 5Monash Newborn, Monash Children’s Hospital, Melbourne, VIC 3168, Australia

**Keywords:** brain injury, cerebral palsy, fetal blood, hypoxic ischemic encephalopathy, infant, intraventricular haemorrhage, newborn

## Abstract

Perinatal brain injury is a major contributor to long-term adverse neurodevelopment. There is mounting preclinical evidence for use of umbilical cord blood (UCB)-derived cell therapy as potential treatment. To systematically review and analyse effects of UCB-derived cell therapy on brain outcomes in preclinical models of perinatal brain injury. MEDLINE and Embase databases were searched for relevant studies. Brain injury outcomes were extracted for meta-analysis to calculate standard mean difference (SMD) with 95% confidence interval (CI), using an inverse variance, random effects model. Outcomes were separated based on grey matter (GM) and white matter (WM) regions where applicable. Risk of bias was assessed using SYRCLE, and GRADE was used to summarise certainty of evidence. Fifty-five eligible studies were included (7 large, 48 small animal models). UCB-derived cell therapy significantly improved outcomes across multiple domains, including decreased infarct size (SMD 0.53; 95% CI (0.32, 0.74), *p* < 0.00001), apoptosis (WM, SMD 1.59; 95%CI (0.86, 2.32), *p* < 0.0001), astrogliosis (GM, SMD 0.56; 95% CI (0.12, 1.01), *p* = 0.01), microglial activation (WM, SMD 1.03; 95% CI (0.40, 1.66), *p* = 0.001), neuroinflammation (TNF-α, SMD 0.84; 95%CI (0.44, 1.25), *p* < 0.0001); as well as improved neuron number (SMD 0.86; 95% CI (0.39, 1.33), *p* = 0.0003), oligodendrocyte number (GM, SMD 3.35; 95 %CI (1.00, 5.69), *p* = 0.005) and motor function (cylinder test, SMD 0.49; 95 %CI (0.23, 0.76), *p* = 0.0003). Risk of bias was determined as serious, and overall certainty of evidence was low. UCB-derived cell therapy is an efficacious treatment in pre-clinical models of perinatal brain injury, however findings are limited by low certainty of evidence.

## SIGNIFICANCE STATEMENT

Perinatal brain injury can lead to significant long-term neurodevelopmental deficits. There are limited treatment options available, and new interventions are urgently required. Through assessment of preclinical studies, this systematic review and meta-analysis shows that umbilical cord blood-derived cell therapy is an efficacious treatment for perinatal brain injury across a wide range of neuropathological and functional domains, albeit with low certainty of evidence. It also identified knowledge gaps, including that future studies should focus on non-hypoxic ischemic models, preterm models, large animal models and should explore the heterogeneity that exists in treatment protocols. Thus, this study stands as a significant contribution to future research direction and the translation of umbilical cord blood-derived cell therapies to human neonates.

## 1. Introduction

Perinatal brain injury refers to a pathological insult to the developing brain either before or during birth, or in the early neonatal period, and carries significant implications for both acute and chronic neurodevelopmental impairment of the child [1]. Perinatal brain injury associated with prematurity and term neonatal encephalopathy continue to be most significant [2]. Proposed mechanisms of prematurity-related brain injury include ischemia secondary to haemorrhage, placental hypoperfusion or fetal vascular dysfunction, as well as inflammation secondary to conditions such as chorioamnionitis, maternal systemic infection and neonatal sepsis [3]. Additionally, neonatal encephalopathy is most commonly caused by hypoxia ischemia, termed hypoxic-ischemic encephalopathy (HIE), secondary to placental abruption, cord prolapse or uterine rupture [4]. However, other pathologies such as perinatal stroke, infection, maternal toxins, fetal growth restriction and intracerebral haemorrhage secondary to birth trauma are also major contributors to the development of neonatal encephalopathy [4]. Together, these forms of perinatal brain injury represent significant risk factors for long-term neurodevelopmental impairment including cerebral palsy, learning and behavioural difficulties, and other neurosensory impairments leading to decreased quality of life with other related medical complications [5,6].

Current interventions for treatment of perinatal brain injuries are limited, but include administration of antenatal steroids and magnesium sulphate during preterm labour, and therapeutic hypothermia in the setting of term HIE [7,8]. However, these treatments are met with challenges and are not appropriate in all settings. For instance, it can be difficult for mothers to receive antenatal interventions, and therapeutic hypothermia is only indicated for term babies within the first 6 h post birth [8,9,10]. Furthermore, although hypothermia reduces the incidence of adverse outcomes, [11] it does not address the underlying brain injury with a substantial number of these infants still experiencing long-term morbidity [12]. Hence, it is evident that there is a pressing need for additional early intervention treatments for perinatal brain injury that is effective, reduces underlying brain injury, and can be applied across multiple preterm and term indications, where appropriate.

One area of promising research is cell therapy, which involves the use of multipotent stem cells and other cells that may possess the ability to self-renew and differentiate into a number of cell lineages [13,14]. Cell therapies can exert their therapeutic effects through multiple proposed mechanisms including, activation of anti-inflammatory and anti-apoptotic molecular pathways, and pro-angiogenic and antioxidant effects [13,14,15]. Human umbilical cord blood (UCB) is acknowledged as a plentiful source of mononuclear cells with multipotency properties, namely mesenchymal stem/stromal cells (MSC), endothelial progenitor cells (EPC) and haematopoietic stem cells (HSC), as well as immunomodulatory cells including T-regulatory cells (Treg) [16,17,18,19]. The use of UCB-derived cell therapy for perinatal brain injury holds many advantages over current existing management. Apart from being an easily accessible source of stem cells from gestational tissue that is often discarded, UCB-derived cell therapy has minimal ethical issues with cell collection, low immunogenicity and low tumorigenicity [20]. Indeed, in early clinical studies, there are promising results suggesting that UCB-derived cell therapy is both safe and feasible [20,21]. However, it remains unclear whether UCB-derived cell therapy is efficacious in the treatment of perinatal brain injury.

We conducted a systematic review and meta-analysis to assess the evidence from preclinical studies on the efficacy of UCB-derived cell therapy in the management of perinatal brain injury. The primary aim was to assess the effects of UCB-derived cell therapy on brain injury outcomes (i.e., infarct size, neuron and oligodendrocyte number, apoptosis, astrogliosis, microglial activation, neuroinflammation and motor function) in preclinical animal models of perinatal brain injury and to identify knowledge gaps in current research.

## 2. Results

### 2.1. Search Results

A flow diagram of study selection is presented in Figure 1, using the PRISMA flowchart template, as previously described [22,23]. The search yielded a total of 1082 citations. A total of 368 duplicates were excluded, with 627 papers excluded at title and abstract screening, and 15 excluded for full-text articles not being able to be retrieved. 72 papers underwent full-text screening, with a further 19 papers excluded for reasons outlined in Figure 1. An additional 2 papers were found upon citation searching of literature. Thus, a total of 55 papers were included in this systematic review and meta-analysis [14,15,24,25,26,27,28,29,30,31,32,33,34,35,36,37,38,39,40,41,42,43,44,45,46,47,48,49,50,51,52,53,54,55,56,57,58,59,60,61,62,63,64,65,66,67,68,69,70,71,72,73,74,75,76].

### 2.2. Characteristics of Included Studies

The characteristics of included studies are summarised in Table 1, as previously described [23]. The most common animal model studied were rats (67%, *n* = 37), followed by mice (15%, *n* = 8), sheep (13%, *n* = 7) and rabbits (6%, *n* = 3). The most common model of brain injury investigated were HI (74%, *n* = 41), followed by IVH (13%, *n* = 7), ischaemic stroke (2%, *n* = 1), chorioamnionitis (3%, *n* = 2), meningitis (2%, *n* = 1), FGR (2%, *n* = 1), hyperoxia (2%, *n* = 1) and excitotoxic brain lesions (2%, *n* = 1). The timing of brain injury ranged from in utero to PND14, whilst the timing of UCB-derived cell therapy administration ranged from 1 h to 7 days post brain injury. Additionally, studies were predominantly term (62%, *n* = 34) and small animal models – rat, mice, or rabbit (87%, *n* = 48).

### 2.3. Markers of Brain Injury

Data for the most common markers used to measure each form of brain injury outcome across studies, were extracted for meta-analysis. The following markers and subsequent data extraction conducted in this review were: tissue and volume loss as a measurement of infarct size, neuronal nuclear protein (NeuN) as a marker for neuron number, myelin basic protein (MBP) as a marker for oligodendrocyte number, terminal deoxynucleotidyl transferase dUTP nick end labelling (TUNEL) and caspase 3 as markers for apoptosis, glial fibrillary acidic protein (GFAP) as a marker for astrogliosis, Iba-1 and ED-1 as markers for microglial activation, TNF-α, IL-6, IL-1 β as markers for neuroinflammation, IL-10 as a marker for anti-inflammation, and cylinder tests and rotarod tests as markers for motor function. For microglial activation Iba-1 (*n* = 12) was prioritised over ED-1 (*n* = 10), and for apoptosis caspase 3 (*n* = 13) was prioritised over TUNEL (*n* = 13).

### 2.4. Effect of Ucb-Derived Cell Therapy on Infarct Size, Neuron Number, Oligodendrocyte Number & Apoptosis

Twenty-three studies assessed infarct size using measurements including tissue loss (*n* = 7), volume loss (*n* = 6) and ipsilateral/contralateral volume ratio (*n* = 10). Nine studies were excluded from the meta-analysis due to unavailability of data or healthy control group comparator [30,36,39,41,42,55,63,67,74]. Of the remaining 14 studies included in the meta-analysis, three had multiple interventional groups based on dose, two had two experimental groups based on sex and one had multiple groups based on UCB cell type, resulting in 28 study entries. Our meta-analysis demonstrated that UCB-derived cell therapy significantly decreased infarct size by a SMD of 0.53 (95% CI 0.32, 0.74; *p* < 0.00001) (Figure 2A).

Ten studies measured NeuN as a marker for neuron number in grey matter structures. Of these 10 studies, one paper had two treatment arms with differing numbers of doses, one had two groups separated based on sex, and one study analysed two types of UCB-derived cells; resulting in 13 study entries. [40,61,62] As seen in Figure 2B, 4 studies had a significant increase in neuron number, with meta-analysis demonstrating that UCB-derived cell therapy significantly increased neuron number by a SMD of 0.86 (95% CI 0.39, 1.33; *p* < 0.001). There were insufficient papers for analysis assessing NeuN in white matter structures.

Eleven studies assessed oligodendrocyte numbers using the marker of MBP. One study was excluded from the meta-analysis due to unavailability of data. [68] Of the remaining ten studies, two assessed MBP in grey matter regions and eight in white matter regions. As seen in Figure 2C, the meta-analysis showed that across three study entries UCB-derived cell therapy significantly increases oligodendrocyte number in grey matter structures (SMD 3.35; 95% CI 1.00, 5.69; *p* < 0.005). Two of the seven studies assessing white matter structures included multiple treatment arms, resulting in a total of 9 study entries. As seen in Figure 2D, UCB-derived cell therapy was associated with a significant increase in oligodendrocyte number in white matter structures (SMD 0.53; 95% CI 0.09, 0.96, *p* = 0.02).

Twenty-five studies evaluated apoptosis using the markers of caspase 3 (*n* = 13) or TUNEL (*n* = 13), with one study evaluating both. [46] Seven studies were excluded from the meta-analysis due to lack of quantitative data available or brain structure not classified as grey or white matter [44,45,53,56,65,68,75]. As evident in Figure 2E,F, the 18 studies included in the meta-analysis were further grouped into studies that assessed grey matter and white matter brain regions. After accounting for multiple treatment groups, across 19 study entries, UCB-derived cell therapy significantly decreased apoptosis in grey matter structures (SMD 0.85; 95% CI 0.37, 1.33; *p* = 0.0005). Similarly, 14 study entries assessing white matter structures showed that UCB-derived cell therapy significantly decreased apoptosis by a SMD of 1.59 (95% CI 0.86, 2.32, *p* < 0.0001).

### 2.5. Effect of Ucb-Derived Cell Therapy on Astrogliosis & Microglia

Twenty six studies assessed astrogliosis using the marker of GFAP, and 9 studies were excluded from the meta-analysis due to lack of available data or the brain structure assessed unable to be classified as either grey or white matter [45,49,56,63,64,68,71,72,74]. Of the remaining 17 studies, a total of 20 study entries assessing grey matter and 20 study entries assessing white matter were included in the meta-analysis. As demonstrated in Figure 3A,B six study entries found a significant improvement in GFAP between control and experimental groups in grey matter, while 8 study entries showed a significant improvement in white matter following UCB-derived cell therapy. In both grey and white matter brain regions, the meta-analysis showed that UCB-derived cell therapy significantly improved astrogliosis (grey matter SMD 0.56; 95% CI 0.12, 1.01; *p* = 0.01), (white matter SMD 0.77; 95% CI 0.22, 1.33; *p* = 0.006).

Twenty-two papers assessed Iba-1 (*n* = 12) or ED-1 (*n* = 10) as a marker of microglial activation in grey matter brain regions and 6 papers assessed this in white matter brain regions. Due to unavailable data for analysis or brain regions not being classified as grey or white matter, 6 papers were excluded for meta-analysis for microglial activation in grey matter [24,30,43,53,62,68], and no papers excluded for white matter. As shown in Figure 3C,D, there were 23 study entries and 8 study entries in the forest plots, respectively. Meta-analysis showed that UCB-derived cell therapy significantly reduced microglial activation in grey matter regions by a SMD of 0.70 (95% CI 0.37, 1.02; *p* < 0.0001). Similarly, UCB-derived cell therapy significantly reduced microglia activation in white matter brain regions, with a SMD of 1.03 (95% CI 0.40. 1.66; *p* = 0.001).

### 2.6. Effect of Ucb-Derived Cell Therapy on Neuroinflammation

Fifteen studies measured TNF-α as a marker of neuroinflammation. One study was excluded due to having no available data [49], with a total of 17 study entries included in our meta-analysis due to 3 papers including multiple treatment arms [25,50,57]. Meta-analysis demonstrated that UCB-derived cell therapy significantly reduced TNF-α with a SMD of 0.84 (95% CI 0.44, 1.25; *p* < 0.0001), Figure 4A. Eleven studies measured IL-6 with 2 studies excluded for having unavailable data for extraction. [49,51] Meta-analysis demonstrated that UCB-derived cell therapy was able to significantly reduce IL-6 by a SMD of 1.05 (95% CI 0.32, 1.79; *p* < 0.01), Figure 4B. Twelve studies measured IL-1β, with meta-analysis showing that UCB-derived cell therapy significantly reduces IL-1β as compared to brain injury controls, by a SMD of 1.11 (95% CI 0.45, 1.77; *p* = 0.001), Figure 4C.

Five studies measured IL-10 as an anti-inflammatory marker, with a total of 1 study excluded from meta-analysis due to unavailable data. [51] Only 1 out of 4 studies demonstrated that UCB-derived cell therapy significantly increased IL-10, with meta-analysis showing no significant differences between UCB-derived cell therapy and injury control groups, as shown in Figure 4D.

### 2.7. Effect of Ucb-Derived Cell Therapy on Motor Function

Eleven studies assessed motor function using the cylinder test however two studies were excluded due to unavailable data [30,35]. Of the remaining nine studies, 3 included multiple treatment groups, resulting in a total of 12 entries to be included in the meta-analysis. Meta-analysis showed that UCB-derived cell therapy significantly improved motor function when compared to the injury control group (SMD 0.49; 95% CI 0.23, 0.76; *p* = 0.0003), Figure 5A. Additionally, 8 out of 55 studies used the rotarod test to measure motor function. Three of these studies were excluded due to inability to confirm quantitative data, resulting in a total of five papers included in the meta-analysis [25,41,67]. UCB-derived cell therapy significantly improved rotarod test ability with a SMD of 1.27 (95% CI 0.45, 2.09; *p* = 0.002), Figure 5B.

### 2.8. Quality Assessment

The risk of bias of included studies is summarised in Figure 6, as previously described [23]. No studies were assessed as low risk of bias across all domains. Selection bias was judged as low across the majority of studies for sequence generation, with 36 studies reporting randomised allocation to experimental groups. However, few studies stated the method of randomisation. Additionally, few studies reported baseline characteristics and allocation concealment presence, resulting in unclear risk of selection bias across these domains. Similarly, performance bias was judged as unclear for nearly all studies, with six studies reporting the blinding of caregivers and one study reporting randomisation of animal housing. In 35 studies, outcome assessors were blinded but only one study reported the randomisation of outcomes, resulting in an unclear risk of detection bias across most studies. Additionally, attrition bias was judged as unclear for all studies. Across all studies, no study protocol was available, resulting in unclear risk of reporting bias. No additional sources of biases were identified such as industry funding or conflict of interest.

As per the GRADE tool guidelines adapted for preclinical studies, initial quality of certainty was “high” due to included studies being randomized trials [77,78]. Across the domains used in the GRADE tool, “risk of bias” was determined to be “serious” due to reasons aforementioned, “imprecision” and “indirectness” as “not serious”, “inconsistency” as “serious” due to the majority of outcomes having moderate-high heterogeneity, and “publication bias” as “serious” due to generated funnel plots detecting asymmetry amongst papers as shown in Appendix A. Certainty was upgraded due to findings being consistent across different species. Thus, the overall certainty of evidence for our findings was low.

## 3. Discussion

There is growing interest in UCB-derived cell therapy for the treatment of perinatal brain injury, with mounting preclinical evidence supporting its efficacy across a range of models and indications. Our systematic review and meta-analysis identified 55 relevant preclinical studies and demonstrated that UCB-derived cell therapy is efficacious, with improvements in outcomes across a wide range of neuropathological, biochemical and functional parameters. It is however important to take these findings in the context of our quality assessment, which found overall low certainty of evidence.

### 3.1. Effect of Ucb-Derived Cell Therapy on Brain Outcomes of Perinatal Brain Injury

As aforementioned, perinatal brain injury associated with HIE and preterm white matter injury are of high clinical relevance [2]. HIE occurs via hypoxia ischemia with subsequent excitotoxicity and neuroinflammation leading to apoptotic cell death and neuronal cell necrosis [79]. Our meta-analyses demonstrated that UCB-derived cell therapy significantly attenuates apoptosis and neuroinflammation, whilst increasing neuron and oligodendrocyte number when compared to controls. Our results also suggest that UCB-derived cell therapy is able to exert its neuromodulatory effects by significantly decreasing glial activation, namely astrogliosis and microglial activation, decreasing infarct size and ultimately resulting in a significant improvement in long-term motor function. This is of particular relevance given the mechanisms underlying brain injury as well as the associated long-term cognitive and motor deficits [5,79]. Furthermore, these findings are consistent with other similar studies in the research field, however there was still significant risk of bias, increased heterogeneity and overall low certainty in results [80,81].

To our knowledge this is the first systematic review and meta-analysis assessing the effect of UCB-derived cell therapy in the treatment of perinatal brain injury across a range of preclinical models. Serrenho et al., (2021) performed a systematic review and meta-analyses of 58 preclinical studies of HIE animal models and similarly concluded that stem-cell therapy may exert its neuroprotective effects via a wide range of mechanisms including promotion of neuronal proliferation, neurogenesis, angiogenesis and inhibition of inflammatory cytokines, apoptosis, astrogliosis and microglial activation [80]. These findings are consistent with the results of our study, however notably in our systematic review, we were also able to synthesise and comment on the efficacy of cell therapy across multiple animal models of brain injury such as IVH, meningitis or FGR [15,26,27].

Additionally, Archambault et al., (2017) conducted a systematic review and meta-analysis looking at the effect of MSCs in preclinical models of HIE that suggested that MSCs are effective in improving both cognitive and functional outcomes [81]. Interestingly, they found that motor function as measured via the cylinder test and rotarod test was improved by a SMD of 2.25 and 2.97, respectively. This is in keeping with the results of our meta-analysis which also found a positive effect, although of smaller efficacy with SMD 0.49 and 1.27, respectively. This noted difference may be due to a number of reasons including that of increased heterogeneity and differences in study design. Archambault et al., (2017) identified significant heterogeneity in their assessment of both the cylinder test and rotarod test, with *I*^2^ = 95.2% and *I*^2^ =85.9%, respectively. Our study on the other hand had a heterogeneity of *I*^2^ =20% and *I*^2^ = 83%, respectively, which may account for the attenuated effect. However, it is also noted that the paper of Archambault et al. (2017) paper included only MSCs, whilst our review included a wide range of additional UCB-derived cells such as EPCs, Tregs and MNCs which have also been shown to be neuroprotective [14]. This may be suggestive that MSCs are more effective than other forms of UCB-derived cell therapy, however interestingly a study performed by McDonald et al., (2018) suggested that EPCs may be the more beneficial than other forms of mononuclear cells [14]. Future studies exploring these differences would be beneficial to the existing literature.

The current literature is also suggestive that UCB-derived cell therapy is effective in the treatment of perinatal brain injury in both small and large animal models. Chang et al., (2021) assessed the efficacy of human UCB-derived CD34+ cells on a mouse model of HIE and showed that these cells were effective in reducing neuronal loss and improving motor function [31]. The neuroprotective effects of UCB-derived cells have also been shown to be effective in larger animal models, with cord blood mononuclear cells being associated with significantly reduced neuronal apoptosis, astrogliosis and inflammation in lamb models of perinatal asphyxia [28]. These findings are consistent with the findings of our systematic review and meta-analysis. Taken together, these findings in preclinical models are suggestive that UCB-derived cell therapy is a promising neuroprotective treatment for a number of perinatal brain injury outcomes.

Interestingly, Dalous (2013) found that UCB-derived mononuclear cells were able to promote anti-inflammatory response measured by IL-10 in rat models of HI, whereas our meta-analysis suggested that overall there was no significant anti-inflammatory effect in the brain [34]. Our findings are in line with other studies such as Aridas (2016) who also found that whilst UCB-derived mononuclear cells were able to reduce neuroinflammation, they did not significantly increase anti-inflammatory markers in lamb models of perinatal asphyxia [28]. These differences may be attributable to differences in protocol and animal model, however further research into the anti-inflammatory properties of UCB-derived cell therapy would be beneficial to elucidating this discrepancy, as here we only synthesised results from IL-10.

### 3.2. Knowledge Gaps Identified

Our systematic review identified a number of knowledge gaps in the current literature that future research should aim to address. We identified an overwhelming focus on HI in animal models, with 41 out of 55 included papers studying HI injury. Although HI is a significant contributor to perinatal brain injury, it is important to explore the effects of UCB-derived cell therapy in the treatment of other brain injury models, given the promising neuroprotective results seen in non-HIE models, and the significant contribution of non-HI perinatal brain injury to long term neurodisability. Indeed, it has been found that UCB-derived cell therapy is able to exert positive neuroprotective effects in mouse models of meningitis [26], rat models of IVH [27] and lamb models of chorioamnionitis. [58]

Similarly, we also identified a predominance of term perinatal brain injury models (*n* = 34 studies) as compared to preterm brain injury models (*n* = 21 studies). This is of particular relevance as the incidence of term HIE has decreased due to improved obstetric care, whilst the incidence of prematurity is increasing despite this, leading to an increasing population of extremely preterm infants at risk of brain injury [3,82]. Hence, with prematurity being the leading cause of perinatal brain injury and accounting for approximately 11% of all live births worldwide, associated with significant risk factors (low birth weight and fetal growth restriction) and non-HI brain injuries such as neonatal infection, IVH and periventricular leukomalacia, it would be beneficial for future studies to further elucidate the efficacy of UCB-derived cell therapy in preterm animal models [3,82].

Additionally, we found that 48 out of 55 papers used small animal models of perinatal brain injury as compared to large animal models. This is possibly due to the fact that studies in large animal models are more costly, time-consuming and logistically demanding, however it is crucial for future studies to pursue larger animal model studies where appropriate, given the abundance of small animal models, as well as the increased value they bring to translation of therapy to humans [83]. Taken together, it would be beneficial for future studies to further assess the use of UCB-derived cell therapy in the treatment of non-HI brain injury, preterm models and large animal models of perinatal brain injury.

### 3.3. Limitations

As with all studies, our systematic review and meta-analysis carries with it a number of limitations. Our meta-analysis was limited by the quality of reported results and subsequent availability of data for analysis. There were a number of instances where studies were unable to be included in meta-analyses due to the lack of sufficient data. Additionally, our systematic review and meta-analysis primarily focused on cells derived from umbilical cord blood. We acknowledge that there are a number of other cell therapies derived from other sources that have been suggested to be efficacious in the treatment of perinatal brain injury, such as neural stem cells, MSCs and EPCs derived from bone marrow, umbilical cord tissue including Wharton Jelly and other placental tissues [80,84,85]. We also acknowledge that the use of stem cells in combination with other therapies such as therapeutic hypothermia may confer increased protection [86]. Synthesising other cell therapies and combination therapies was outside the scope of this review.

Another limitation of this study was the significant heterogeneity of study design and outcomes across a wide range of domains. This included differences in animal model type, UCB cell type used, timing and route of cell administration, and dosages. Additionally, there was significant heterogeneity in the methodology surrounding collection of data, including type of test used and timing of data collection. For example, neuroinflammation as measured through TNF-α was measured latest at 24 h in one study [15] vs. 28 days in another [27]. It is important for future studies to standardize methodology for more effective comparison of results with consideration of relevant underlying pathological processes. Given the wide range of outcome types, we also acknowledge that there were a number of outcome measures that were not included in this review where UCB-derived cell therapy may also be of benefit, including cognition, neurogenesis and angiogenesis [27,87]. Due to the widespread heterogeneity across studies, direct comparison between studies was difficult, especially in the context of limited available literature delineating the effects of each factor on brain injury outcomes, thus caution should be taken when interpreting our results.

Through our risk of bias assessment using the SYRCLE tool, we were able to identify that a significant number of studies included in our systematic review and meta-analysis had poor reporting across a wide range of domains including baseline characteristics, allocation concealment and randomisation of outcome assessment. This limits the strength of conclusions able to be drawn from our meta-analysis. Unfortunately, this limitation is commonly reported across preclinical animal studies [88]. Taken together with the presence of publication bias on funnel plot assessment, the level of certainty of our findings were determined as low as per the GRADE tool. We suggest that future studies use risk of bias tools such as SYRCLE when reporting methods and results, as well as to publish research protocol before commencement of research to overcome these limitations [89].

### 3.4. Future Directions

At present there are a number of clinical trials underway in humans, with 11 currently registered trials assessing UCB-derived and umbilical cord-derived cells in infants with HIE and preterm brain injury [21]. For example, a completed study in 2020 administered UCB-derived cell therapy to 6 term human newborns with HIE, and found that the intervention was both safe and feasible [20]. Additionally, there are similar trials underway such as that of Malhotra et al. (2020), that seeks to do the same in extremely preterm (<28 weeks gestation) newborns [90]. Given the positive evidence shown in preclinical studies, UCB-derived cell therapy presents itself as a promising and emerging neuroprotective therapy.

## 4. Methods

This systematic review and meta-analysis utilised the Preferred Reporting Items for Systematic Reviews and Meta-Analyses (PRISMA) guidelines [22], and the research protocol was registered on PROSPERO (CRD42022275764).

### 4.1. Selection Criteria

Studies included in our systematic review were preclinical animal studies of any design. Studies were eligible for selection regardless of date of publication or language but were only considered if the full article was available for analysis. Study selection criteria were: (1) any animal model, (2) perinatal brain injury including: low birth weight, fetal growth restriction, placental abnormalities, ischemia, inflammation, hypoxia, asphyxia, intraventricular haemorrhage, periventricular haemorrhage, toxins, intracerebral haemorrhage or trauma, (3) umbilical cord blood-derived cell interventions including: all mononuclear cells, mesenchymal stem/stromal cells, endothelial progenitor cells, hematopoietic stem cells, T-regulatory cells or unrestricted somatic stem cells, and (4) brain structural and functional outcomes including: infarct size, neuron and oligodendrocyte number, apoptosis, astrogliosis, microglial activation, neuroinflammation and motor function. Studies must have compared the effects of umbilical cord blood-derived cells to no intervention, placebo or other stem cells. Additionally, studies were included regardless of whether umbilical cord blood cells were derived from humans or animals, route of administration and timing of intervention.

Studies were excluded if they assessed non-perinatal brain injury models or used adult models. Studies that used non-UCB-derived cells (i.e., derived from adult tissue, placental tissue, bone marrow or umbilical cord tissue), or assessed the efficacy of cells in combination with other interventions were also excluded. Additionally, in vitro studies, case studies and studies that were not primary research (i.e., systematic reviews, literature reviews) were excluded.

### 4.2. Search Strategy

MEDLINE and Embase databases were searched for eligible studies in June 2021, April 2022 and August 2022, by authors TN & EP. The complete search strategy can be found in Appendix A, as previously also described [23].

### 4.3. Study Selection

Studies were exported into Covidence Systematic Review Software (Veritas Health Innovation, Melbourne, Australia, available at www.covidence.org). Duplicates were automatically removed, and a preliminary study title and abstract screen was independently performed by two reviewers (TN & EP). Full texts of potential eligible studies were then independently assessed by two reviewers (TN & EP). Any disagreements throughout the study selection process were resolved through discussion with a third reviewer.

### 4.4. Data Extraction

Relevant data were extracted from eligible studies by two study authors (TN & EP). Data extracted included author name, publication year, animal species, type of animal model, type of perinatal brain injury, control group details and age of injury induction. Additionally, intervention details extracted included cell type, origin species, route of administration, timing of administration, cell dosage and the number of doses given. PlotDigitizer (version 2.6.9) was used to extract quantitative data, for relevant outcomes, from figures when standardised mean difference and standard error or standard deviation was not provided as text or in tables. For papers where data was not readily available, corresponding authors were contacted a total of three times via email, and if authors did not provide a response, the paper was not included for that particular outcome.

Given the expected heterogeneity of brain injury outcome measurements between studies, data was extracted for brain injury markers that were most commonly used amongst the included papers. Brain injury outcomes that were assessed were: infarct size, neuron and oligodendrocyte number, apoptosis, astrogliosis, microglial activation, neuroinflammation and motor function.

### 4.5. Data Synthesis and Statistical Analysis

Quantitative data extracted for primary outcomes were analysed using Review Manager (RevMan) version 5.4. A random-effects, inverse variance model was used to calculate standardized mean difference (SMD) and 95% confidence interval (CI) for continuous data. A SMD of 0.2 represents a small effect, 0.5 represents a medium effect and 0.8 or larger represents a large effect. Heterogeneity was assessed using the *I*^2^ statistic, which is used to measure variation that is due to the heterogeneity across studies as opposed to chance, with 25% considered low, 50% considered moderate and 75% considered high heterogeneity.

For papers that had multiple relevant distinct treatment arms, each treatment arm was included as a separate entry in that forest plot. For outcomes that were measured at multiple timepoints, the final timepoint was utilised in data analysis. Outcomes were separated into grey matter and white matter brain regions when possible. However, in instances where multiple types of grey and white matter regions were assessed in the one study, brain regions of higher frequency of assessment across studies were prioritised for data extraction. A similar approach was taken with neuroinflammatory markers when studies assessed levels in both plasma and tissue. Papers that did not specify brain region of assessment as grey or white matter were excluded from analysis. For brain injury outcomes with multiple markers, a hierarchy was developed based on highest frequency across studies.

### 4.6. Quality Assessment

The Systematic Review Centre for Laboratory Animal Experimentation (SYRCLE) risk of bias tool was used to independently assess the risk of bias of all papers by two reviewers (TN & EP), with all discrepancies resolved through discussion with additional authors, as previously described [23,89]. Additionally, funnel plot analysis including Egger’s test was performed to assess the presence of publication bias using MedCalc for Windows, v20.115 (MedCalc Software, Ostend, Belgium), with certainty of evidence assessed using the Grading of Recommendations Assessment, Development and Evaluation (GRADE) tool adapted for preclinical studies [77].

## 5. Conclusions

This systematic review and meta-analysis of preclinical studies demonstrates that UCB-derived cell therapy is a promising intervention for perinatal brain injury models through its neuroprotective effects across a wide range of neuropathological and functional domains, albeit with low certainty. We have also been able to identify areas of research that warrant further assessment including more studies of non-HIE models, preterm preclinical models and large animal models.

## Figures and Tables

**Figure 1 ijms-24-04351-f001:**
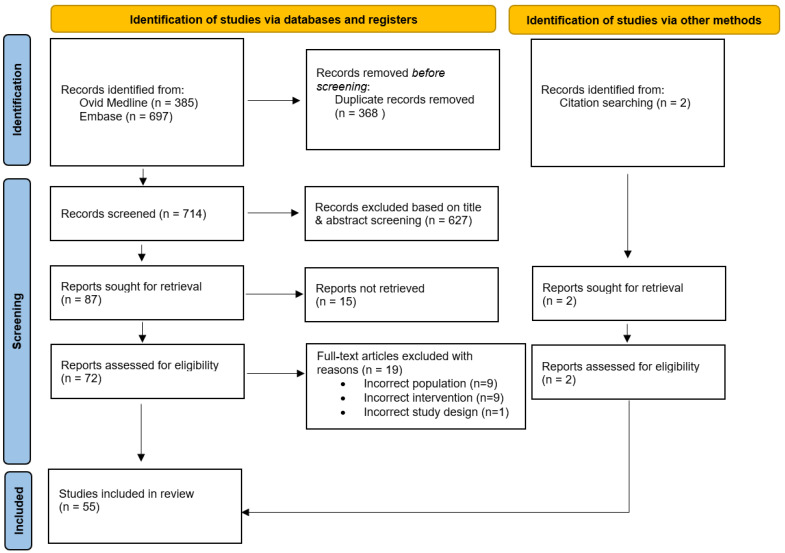
PRISMA diagram detailing study selection process.

**Figure 2 ijms-24-04351-f002:**
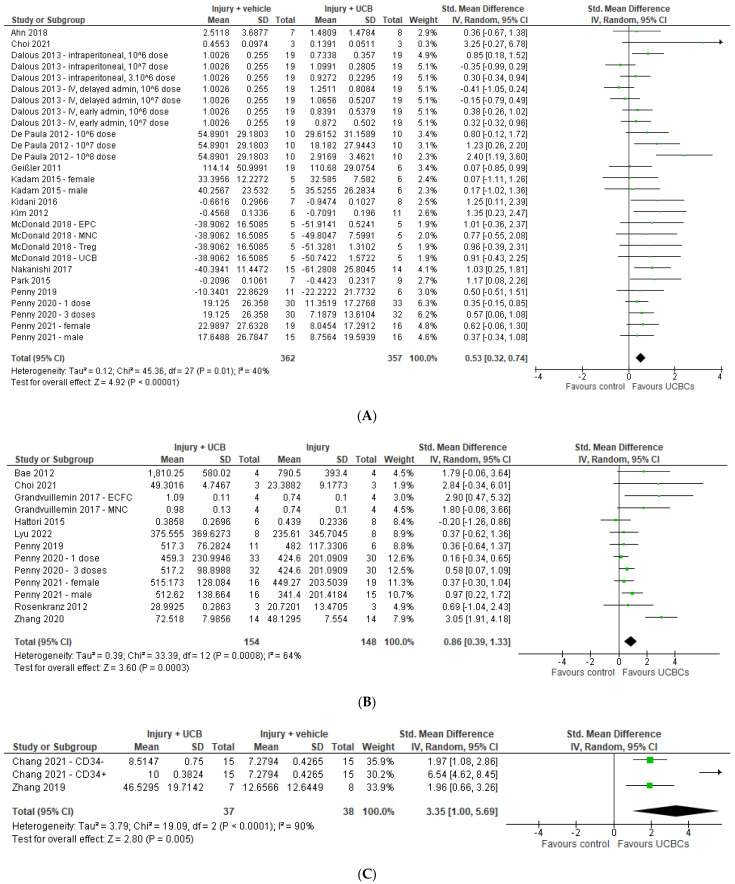
Forest plot demonstrating the effect of umbilical cord blood-derived cell therapy on (**A**) Infarct size; (**B**) Neuron number; (**C**) Oligodendrocyte number—grey matter; (**D**) Oligodendrocyte number—white matter; (**E**) Apoptosis—grey matter; (**F**) Apoptosis—white matter. Umbilical cord blood-derived cell therapy significantly decreased infarct size (*p* < 0.00001), increased neuron number (*p* < 0.001), increased oligodendrocyte number in both grey matter (*p* < 0.0005) and white matter (*p* = 0.02), and decreased apoptosis in both grey matter (*p* = 0.0005) and white matter (*p* < 0.0001). Abbreviations: admin, administration; ECFC, endothelial colony forming cells; EPC, endothelial progenitor cell; HSCs, haemopoietic stem cells; ICV, intracerebroventricular; IV, intravenous; MNC, mononuclear cell; PCB, preterm cord blood; TCB, term cord blood; Treg, T-regulatory cells; UCBC, umbilical cord blood cells.

**Figure 3 ijms-24-04351-f003:**
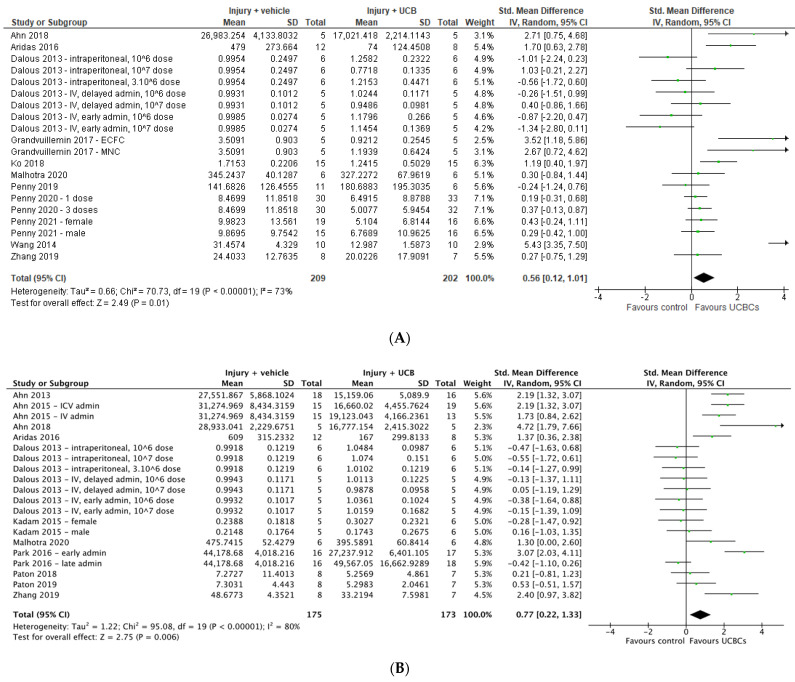
Forest plot demonstrating the effect of umbilical cord blood-derived cell therapy on (**A**) Astrogliosis—grey matter; **(B)** Astrogliosis—white matter; **(C)** Microglia activation—grey matter; **(D)** Microglia activation- white matter. Umbilical cord blood-derived cell therapy significantly decreased astrogliosis in both grey matter (*p* = 0.01) and white matter (*p* = 0.006), and decreased microglia activation in both grey matter (*p* < 0.0001) and white matter (*p* = 0.001).Abbreviations: admin, administration; ECFC, endothelial colony forming cells; EPC, endothelial progenitor cell; ICV, intracerebroventricular; IV, intravenous; MNC, mononuclear cell; PCB, preterm cord blood; TCB, term cord blood; Treg, T-regulatory cells; UCBC, umbilical cord blood cells.

**Figure 4 ijms-24-04351-f004:**
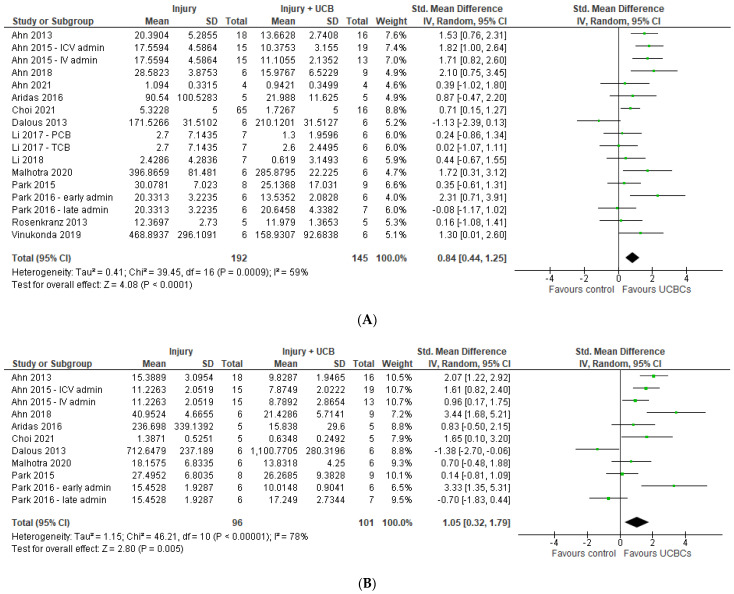
Forest plot demonstrating the effect of umbilical cord blood-derived cell therapy on neuroinflammation (**A**) TNF-α; (**B**) IL-6; (**C**) IL-1β; (**D**) IL-10. Umbilical cord blood-derived cell therapy significantly decreased neuroinflammation as measured by TNF-α (*p* < 0.0001), IL-6 (*p* < 0.01) and IL-1β (*p* = 0.001). Abbreviations: admin, administration; ICV, intracerebroventricular; IV, intravenous; PCB, preterm cord blood; TCB, term cord blood; UCBC, umbilical cord blood cells.

**Figure 5 ijms-24-04351-f005:**
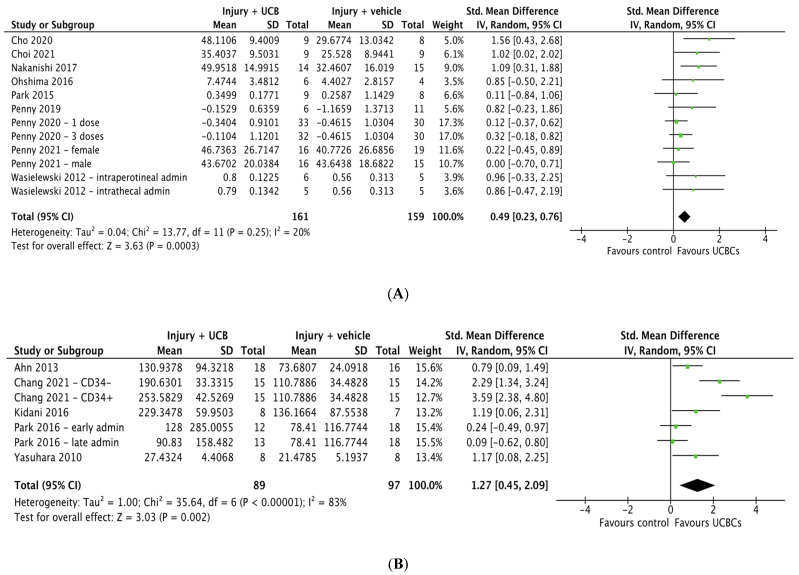
Forest plot demonstrating the effect of umbilical cord blood-derived cell therapy on motor function (**A**) Cylinder test; (**B**) Rotarod test. Umbilical cord blood-derived cell therapy significantly improved motor function as measured by cylinder test (*p* = 0.0003) and rotarod test (*p* = 0.002) Abbreviations: admin, administration; UCBC, umbilical cord blood cells.

**Figure 6 ijms-24-04351-f006:**
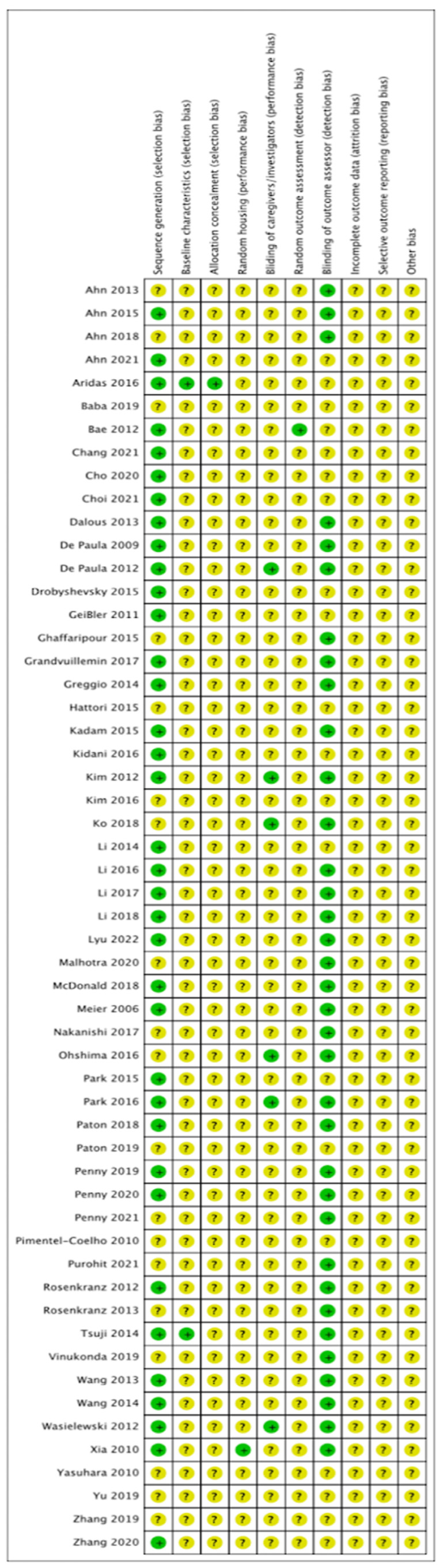
Systematic Review Centre for Laboratory Animal Experimentation (SYRCLE) risk of bias assessment.

**Table 1 ijms-24-04351-t001:** Characteristics of included studies.

Study	Strain and Species	Brain Injury Model	Term vs Pre-term Model	Age Injury Induced	UCB Cell Source & Type	Route of Administration	Total Cells per Dose	Cell Administration Time Post-injury	Comparator
Ahn 2013 [24]	Sprague-Dawley rats	IVH	Pre-term	PND4	Human MSCs	Intraventricular	1 × 10^5^	2 days	Injury + PBS
Ahn 2015 [25]	Sprague-Dawley rats	IVH	Pre-term	PND4	Human MSCs	IC or IV	1 × 10^5^ (IC) or 5 × 10^5^ (IV)	2 days	Injury + NS
Ahn 2018 [26]	Sprague-Dawley rats	Meningitis	Term	PND11	Human MSCs	Intraventricular	1 × 10^5^	6 h	Injury + NS
Ahn 2021 [27]	Sprague-Dawley rats	IVH	Pre-term	PND4	Human MSCs	Intraventricular	1 × 10^5^	2 days	Injury + PBS
Aridas 2016 [28]	Merino Border Leicester cross sheep	HI	Term	139-141 days of gestation	Sheep MNCs	Arterial	1 × 10^8^	12 h after birth	Injury + no vehicle
Baba 2019 [29]	NOD/SCID mice	HI	Term	PND9	Human MNCs	IV	5 × 10^6^	21 days	Injury + PBS
Bae 2012 [30]	Sprague-Dawley rats	HI	Term	PND7	Human MNCs	IV	1 × 10^7^	1 day	Injury + PBS
Chang 2021 [31]	NOD/SCID mice	HI	Term	PND9	Human CD34+ or CD34- HSCs	IC	1 × 10^5^	12 h	Injury + no vehicle
Cho 2020 [32]	ICR mice	HI	Pre-term	PND7	Human MNCs	IP	3 × 10^7^	7 days	Injury + no vehicle
Choi 2021 [33]	ICR mice	HI	Pre-term	PND7	Human MNCs	IP	3 × 10^7^	7 days	Injury + PBS
Dalous 2013 [34]	Sprague-Dawley rats	Excitotoxic brain injury	Pre-term	PND5	Human MNCs	IP or IV	10^6^, 3 × 10^6^ or 10^7^ (IP) 10^6^ or 10^7^ (IV)	1 or 24 h (IP), 6 or 24 h (IV)	Injury + saline
De Paula 2009	Wistar rats	HI	Term	PND7	Human MNCs	IV	1 × 10^7^	1 day	Injury + saline soln
De Paula 2012 [36]	Wistar rats	HI	Term	PND7	Human MNCs	IV	1 × 10^6^, 1 × 10^7^ or 1 × 10^8^	1 day	Injury + vehicle
Drobyshevsky 2015 [37]	New Zealand white rabbits	HI	Pre-term	22 days of gestation	Human MNCs	IV	2.5 × 10^6^ or 5 × 10^6^	4 h after birth	Injury + saline
Geissler 2011 [38]	Wistar rats	HI	Term	PND7	Human MNCs	IP	1 × 10^7^	1 day	Injury + NS
Ghaffaripour 2015 [39]	Wistar rats	HI	Term	PND14	Human MNCs	IV	2 × 10^5^	7 days	Injury + saline soln
Grandvuillemin 2017 [40]	Sprague-Dawley rats	HI	Term	PND7	Human MNCs or ECFCs	IP	1 × 10^7^ (MNC) or 5 × 10^5^ (ECFC)	2 days	Injury + saline soln
Greggio 2014 [41]	Wistar rats	HI	Term	PND7	Human MNCs	Arterial	1 × 10^6^ or 1 × 10^7^	1 day	Injury + vehicle
Hattori 2015 [42]	Wistar rats	HI	Term	PND7	Human MNCs	IP	1 × 10^7^	6 h	Injury + vehicle
Kadam 2015 [43]	CD1 mice	HI	Term	PND12	Human CD34+ enriched MNCs	IP	1 × 10^5^	2 days	Injury + vehicle
Kidani 2016 [44]	SCID mice	HI	Pre-term	PND7	Human CD133+ cells	IP	1 × 10^5^	1 day	Injury + NS
Kim 2012 [45]	Sprague-Dawley rats	HI	Term	PND10	Human MSCs	Intraventricular	1 × 10^5^	6 h	Injury + PBS
Kim 2016 [46]	Sprague-Dawley rats	Hyperoxia	Pre-term	PND0-14	Human MSCs	Intratracheal	1 × 10^5^	PND5	Injury + NS
Ko 2018 [47]	Sprague-Dawley rats	IVH	Pre-term	PND4	Human MSCs	Intracerebroventricular	1 × 10^5^	2 days	Injury + NS
Li 2014 [48]	Sprague-Dawley rats	HI	Pre-term	PND7	Human MNCs or CD34+ cells	IV	1.5 × 10^6^	7 days	Injury + saline
Li 2016 [49]	Merino-Border Leicester cross sheep	HI	Pre-term	102.2 ± 0.3 days of gestation	Sheep MNCs	IV	5 × 10^7^	12 h or 5 days	Injury + saline
Li 2017 [50]	Merino-Border Leicester cross sheep	HI	Pre-term	102.2 ± 0.3 days of gestation	Sheep MNCs	IV	5 × 10^7^	12 h	Injury + saline
Li 2018 [51]	Merino-Border Leicester cross sheep	HI	Pre-term	102.2 ± 0.2 days of gestation	Sheep MNCs	IV	5 × 10^7^	12 h	Injury + saline
Lyu 2022 [52]	Unspecified rats	HI	Term	PND7	Human MNCs	IV	1 × 10^7^	1 day	Injury + NS
Malhotra 2020 [15]	Border Leicester- Merino cross sheep	FGR	Pre-term	88 days of gestation	Sheep MNCs	IV	2.5 × 10^7^	1 h after birth	Injury + saline
McDonald 2018 [14]	Sprague-Dawley rats	HI	Term	PND7	Human MNCs, Tregs, monocytes, EPCs	IP	1 × 10^6^ (MNCs) or 2 × 10^5^ (other)	1 day	Injury + PBS
Meier 2006 [53]	Wistar rats	HI	Term	PND7	Human MNCs	IP	1 × 10^7^	1 day	Injury + NS
Nakanishi 2017 [54]	Sprague-Dawley rats	HI	Term	PND7	Rat MNCs	IP	2 × 10^6^	3 days	Injury + PBS
Ohshima 2016 [55]	CB-17 SCID mice	HI	Pre-term	PND8	Human CD34+ cells	IV	1 × 10^5^	2 days	Injury + PBS
Park 2015 [56]	Sprague-Dawley rats	HI	Pre-term	PND7	Human MSCs	Intraventricular	1 × 10^5^	6 h	Injury + no vehicle
Park 2016 [57]	Sprague-Dawley rats	IVH	Term	PND4	Human MSCs	Intraventricular	1 × 10^5^	2 or 7 days	Injury + PBS
Paton 2018 [58]	Border Leicester-Merino cross sheep	Chorioamnionitis	Pre-term	95 days of gestation	Human MNCs	IV	1 × 10^8^	6 h	Injury + saline
Paton 2019 [59]	Border Leicester-Merino cross sheep	Chorioamnionitis	Pre-term	95 days of gestation	Human MNCs	IV	1 × 10^8^	6 h	Injury + saline
Penny 2019 [60]	Sprague-Dawley rats	HI	Term	PND7	Human MNCs	IP	1 × 10^6^	1 day	Injury + PBS
Penny 2020 [61]	Sprague-Dawley rats	HI	Term	PND10	Human MNCs	Intranasal or IP	1 × 10^6^	1 day (1 dose group) or 1, 3 and 10 days (3 dose group)	Injury + saline
Penny 2021 [62]	Sprague-Dawley rats	HI	Term	PND10	Human MNCs	Intranasal or IP	1 × 10^6^	1, 3 and 10 days	Injury + saline
Pimentel-Coelho 2010 [63]	Lister-Hooded rats	HI	Term	PND7	Human MNCs	IP	2 × 10^6^	3 h	Injury + vehicle
Purohit 2021 [64]	New Zealand white rabbits	IVH	Pre-term	3–4 h after birth	Human unrestricted somatic stem cells	Intraventricular	2 × 10^6^	18 h	Injury + saline
Rosenkranz 2012 [65]	Wistar rats	HI	Term	PND7	Human MNCs	IP	1 × 10^7^	1 day	Injury + vehicle
Rosenkranz 2013 [66]	Wistar rats	HI	Term	PND7	Human MNCs	IP	1 × 10^7^	1 day	Injury + vehicle
Tsuji 2014 [67]	CB-17 SCID mice	Ischaemic stroke	Term	PND12	Human CD34+ cells	IV	1 × 10^5^	2 days	Injury + PBS
Vinukonda 2019 [68]	New Zealand white rabbits	IVH	Pre-term	3-4 h after birth	Human unrestricted somatic stem cells	Intraventricular or IV	2 × 10^6^ (intraventricular) or 1 × 10^6^ (IV)	18 h	Injury + saline
Wang 2013 [69]	Sprague-Dawley rats	HI	Term	PND7	Human MNCs	Intraventricular	3 × 10^6^	1 day	Injury + PBS
Wang 2014 [70]	Sprague-Dawley rats	HI	Term	PND7	Human MNCs	Intraventricular	3 × 10^6^	1 day	Injury + PBS
Wasielewski 2012 [71]	Wistar rats	HI	Term	PND7	Human MNCs	IP or intrathecal	1 × 10^7^	1 day	Injury + saline
Xia 2010 [72]	Sprague-Dawley rats	HI	Term	PND7	Human MSCs	IC	1 × 10^5^	3 days	Injury + vehicle
Yasuhara 2010 [73]	Sprague-Dawley rats	HI	Term	PND7	Human MNCs	IV	1.5 × 10^6^	7 days	Injury + PBS
Yu 2019 [74]	Sprague-Dawley rats	HI	Term	PND7	Human MNCS or CD34+ cells	IV	1 × 10^6^ (MNCs) or 1.5 × 10^4^ (CD34+)	7 days	Injury + PBS
Zhang 2019 [75]	Sprague-Dawley rats	HI	Term	PND7	Human MNCs	Intraventricular	1 × 10^7^	1 day	Injury + PBS
Zhang 2020 [76]	Sprague-Dawley rats	HI	Term	PND7	Human MNCs	Intraventricular	3 × 10^6^	1 day	Injury + NS

Abbreviations: ECFC, endothelial colony forming cells; EPCs, endothelial progenitor cells; FGR, foetal growth restricted; HI, Hypoxia Ischaemia; HSCs, haemopoietic stem cells; IC, intracerebral; ICR, institute for cancer research; IP, intraperitoneal; IV, intravenous; IVH, intraventricular haemorrhage; MNCs, mononuclear cells; MSCs, mesenchymal stem cells; NS, normal saline; NOD, nonobese diabetic; PBS, phosphate buffered saline; PND, post-natal day; SCID, severe combined immunodeficient; soln, solution; Tregs, T regulatory cells.; UCB, umbilical cord blood.

## Data Availability

All datasets and analyses created in this review are available upon reasonable request.

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
