# Peer review of "Umbilical Cord Blood-Derived Cell Therapy for Perinatal Brain Injury: A Systematic Review & Meta-Analysis of Preclinical Studies"

_ijms, 2023, doi:10.3390/ijms24054351_

Round 1

Reviewer 1 Report

The theme of this manuscript is important from both research and clinical points of view. The strength of this review includes 1) various types of brain injuries around perinatal period and 2) all sorts of umbilical cord blood (UCB)-derived cells are analyzed all together.  

Major comments

Overall, this study is well organized. This study, however, has the following weakness:

1) This study put emphasis on morphological and biological assessment, but not behavioral assessment, with respect to the efficacy of UCB-derived cells. Only the results of rotarod and cylinder tests were analyzed, and no cognitive test was analyzed. As cognitive impairments are critical in the quality of life of patients and their family, the assessment of cognitive impairment is important. Evaluations on neurogenesis and angiogenesis may be of some importance as well.

2) This study excludes combination therapies with other treatments. As therapeutic hypothermia became the standard treatment for neonatal hypoxic-ischemic encephalopathy, it would be important to evaluate additive effect of cell therapy over hypothermia. 

3) The data at final timepoint were used for analysis if the outcome was assessed several times. Such evaluation method is appropriate to evaluate the behavioral outcome, but not for some parameters including cytokines, many of which increase during acute phase and decrease within a few days, sometimes in one day. Therefore, the treatment effects on those parameters may be better detected at the earlier time point. 

 Minor comments

1) The reference numbers are wrong in many cases. For instance, a research paper by Meier et al. is described as [53] in the table 1, but the paper is listed as [55] in the Reference.

2) It would be helpful for general readers to explain the meaning of SMD briefly.  

Author Response

Reviewer 1

The theme of this manuscript is important from both research and clinical points of view. The strength of this review includes 1) various types of brain injuries around perinatal period and 2) all sorts of umbilical cord blood (UCB)-derived cells are analyzed all together.  

  • Thank you for your comments

Major comments

Overall, this study is well organized. This study, however, has the following weakness:

1) This study put emphasis on morphological and biological assessment, but not behavioral assessment, with respect to the efficacy of UCB-derived cells. Only the results of rotarod and cylinder tests were analyzed, and no cognitive test was analyzed. As cognitive impairments are critical in the quality of life of patients and their family, the assessment of cognitive impairment is important. Evaluations on neurogenesis and angiogenesis may be of some importance as well.

  • We acknowledge that there are a number of other important outcome parameters where UCB-derived cells may prove to be beneficial, which were not included in this study. We have included this limitation in our discussion, but we limited by what the original studies had studied.

2) This study excludes combination therapies with other treatments. As therapeutic hypothermia became the standard treatment for neonatal hypoxic-ischemic encephalopathy, it would be important to evaluate additive effect of cell therapy over hypothermia. 

  • We acknowledge exploring the combination and additive effect of cell therapy is an important consideration. We have included this limitation in our discussion.

3) The data at final timepoint were used for analysis if the outcome was assessed several times. Such evaluation method is appropriate to evaluate the behavioral outcome, but not for some parameters including cytokines, many of which increase during acute phase and decrease within a few days, sometimes in one day. Therefore, the treatment effects on those parameters may be better detected at the earlier time point. 

  • We have addressed this concern in the limitation section of our discussion. Given the vast heterogeneity of methodology across studies it was difficult to compare results where data collection timepoints differed greatly. The decision to use the final timepoint was collectively made based on similar published literature in the field [1] and for consistency across outcome analysis throughout the paper. However, we acknowledge that this is an important consideration, and have implored future studies to standardize methodology for more effective comparison and to consider underlying pathological processes.

[1] Madeleine J. Smith, Madison Claire Badawy Paton, Michael C. Fahey, Graham Jenkin, Suzanne L. Miller, Megan Finch-Edmondson, Courtney A. McDonald, Neural Stem Cell Treatment for Perinatal Brain Injury: A Systematic Review and Meta-Analysis of Preclinical Studies, Stem Cells Translational Medicine, Volume 10, Issue 12, December 2021, Pages 1621–1636, https://doi.org/10.1002/sctm.21-0243

 Minor comments

1) The reference numbers are wrong in many cases. For instance, a research paper by Meier et al. is described as [53] in the table 1, but the paper is listed as [55] in the Reference.

  • This has now been corrected.

2) It would be helpful for general readers to explain the meaning of SMD briefly.  

  • A brief description of how to interpret SMD has been added to the methods section.

Reviewer 2 Report

The author/s has written a comprehensive systematic review article entitled “Umbilical cord blood (UCB)-derived cell therapy for perinatal brain injury: A systematic review & meta-analysis of preclinical studies” in the field of perinatal brain injury. In this article, the authors cautiously determined that UCB-derived cell therapy appeared to be efficacious in pre-clinical models of perinatal brain injury due to a limited number of evidence. In particular, the underpinning pathways that the UCB-derived cell therapy acts on are apoptosis, astrogliosis, microglial activation, neuroinflammation, oligodendrocyte number, motor function and decrease infarct size.

Weakness:

1.      Abstract section: Please make sure whether subtitles like “introduction”, “objective”, “ Methods”, “Results” and so on are necessary. The author defined the full form of SMD and Cl but I could not the numerical values means inside the parenthesis  “SMD 0.53; 95%CI (0.32, 0.74)” here and elsewhere. In addition, making sure the Significance Statement is important should need to check with the author's Journals guidelines. This significance statement is more fitting in the “Conclusion” section.

2.      Please define I2 statistics and the reason to use them in the study.

3.      Table 1: The title can be given here. But all the abbreviations look better fit if you move below the Table.

4.      The authors also need to consult a professional statistician on how to best represent the figures in a graphical form such as HITMAP or similar for large datasets. All the Tabular form data (termed as Figure herein) just looks like too much information and the reader can easily get distracted losing the main message from the Figures. I also prefer to represent the significant value while making comparisons thorough out the figures. This is one of the major issues in the manuscript.

5.      Please provide more details on the figure legends about the significant findings about each specific figure throughout the manuscript.

The font size and type of text should be consistent especially in the Tables or Figures for clarity.

6.      Statement like “Studies included animal models of rats (67%, n = 37), mice (15%, n = 8), sheep (13%, n = 7) and rabbits (6%, n = 3). The models of brain injury investigated were HI (74%, n = 41), IVH (13%, n = 7), ischaemic stroke (2%, n = 1), chorioamnionitis (3%, n = 2), meningitis (2%, n = 1), FGR (2%, n = 1), hyperoxia (2%, n = 1) and excitotoxic brain lesions (2%, n = 1)”. Wherever you see this type of hanging data in the manuscript, please make sense of the data and interpret its meaning.  In general, comparative discussion can obviously fall in the discussion section. My concern is that the data seems disconnected from the logical flow of the text in some areas as pointed out above and readers may not quite understand this complexity.

Author Response

Reviewer 2

The author/s has written a comprehensive systematic review article entitled “Umbilical cord blood (UCB)-derived cell therapy for perinatal brain injury: A systematic review & meta-analysis of preclinical studies” in the field of perinatal brain injury. In this article, the authors cautiously determined that UCB-derived cell therapy appeared to be efficacious in pre-clinical models of perinatal brain injury due to a limited number of evidence. In particular, the underpinning pathways that the UCB-derived cell therapy acts on are apoptosis, astrogliosis, microglial activation, neuroinflammation, oligodendrocyte number, motor function and decrease infarct size.

  • Thank you for your comments

Weakness:

  1. Abstract section: Please make sure whether subtitles like “introduction”, “objective”, “ Methods”, “Results” and so on are necessary. The author defined the full form of SMD and Cl but I could not the numerical values means inside the parenthesis  “SMD 0.53; 95%CI (0.32, 0.74)” here and elsewhere. In addition, making sure the Significance Statement is important should need to check with the author's Journals guidelines. This significance statement is more fitting in the “Conclusion” section.
  • We believe that the inclusion of listed subtitles makes it easier on the reader to absorb the information presented in the Abstract
  • The numbers in the parenthesis outline the lower and upper range of the 95% confidence interval
  • We have edited the significance statement to better highlight the contributions to the literature
  1. Please define Istatistics and the reason to use them in the study.
  • We have edited the methods section to include an explanation of I2 The relevance of this analysis was to determine to what extent the SMD calculated was influenced by heterogeneity across papers.
  1. Table 1: The title can be given here. But all the abbreviations look better fit if you move below the Table.
  • We have moved the abbreviations to below the Table.
  1. The authors also need to consult a professional statistician on how to best represent the figures in a graphical form such as HITMAP or similar for large datasets. All the Tabular form data (termed as Figure herein) just looks like too much information and the reader can easily get distracted losing the main message from the Figures. I also prefer to represent the significant value while making comparisons thorough out the figures. This is one of the major issues in the manuscript.
  • We thank you for your comment and appreciate the complexity demonstrated in the data. As you later comment, we agree the data is complex and given the scope and magnitude of the review, in both number of studies included and outcomes assessed, there are naturally many figures to display and discuss. Forest plots are still the most common way to display data in systematic reviews, and we are of the belief that to do the data justice, it is appropriate to present it as such. With additions to the figure legends as suggested below highlighting the relevant significant findings, we believe this makes the data much more palatable. This is also in keeping with other systematic reviews and meta-analyses in the literature surrounding similar topics. [1]

[1] Archambault J, Moreira A, McDaniel D, Winter L, Sun L, Hornsby P. Therapeutic potential of mesenchymal stromal cells for hypoxic ischemic encephalopathy: A systematic review and meta-analysis of preclinical studies. PLoS One. 2017 Dec 19;12(12):e0189895. doi: 10.1371/journal.pone.0189895. PMID: 29261798; PMCID: PMC5736208.

  1. Please provide more details on the figure legends about the significant findings about each specific figure throughout the manuscript.
  • We have added this to each figure legend.

The font size and type of text should be consistent especially in the Tables or Figures for clarity

  • We have amended this.
  1. Statement like “Studies included animal models of rats (67%, n = 37), mice (15%, n = 8), sheep (13%, n = 7) and rabbits (6%, n = 3). The models of brain injury investigated were HI (74%, n = 41), IVH (13%, n = 7), ischaemic stroke (2%, n = 1), chorioamnionitis (3%, n = 2), meningitis (2%, n = 1), FGR (2%, n = 1), hyperoxia (2%, n = 1) and excitotoxic brain lesions (2%, n = 1)”. Wherever you see this type of hanging data in the manuscript, please make sense of the data and interpret its meaning.  In general, comparative discussion can obviously fall in the discussion section. My concern is that the data seems disconnected from the logical flow of the text in some areas as pointed out above and readers may not quite understand this complexity.
  • We have edited this section to more appropriately highlight the most common animal model and brain injury model used across the included studies. We believe this fits well as an introduction to the results section as it explains the characteristics of included studies before moving onto analysis of the data and outcomes.

Round 2

Reviewer 2 Report

-Since the authors believe that the Tabular form are better way than graphical representation (bar, chart, hwat map or others) to represent this type of data as few authors have adopted this practice, I will not have further comments on it.

- I also suggest to explain those studies where n=1, in their sample size issues.